# The Challenges and Opportunities in the Development of MicroRNA Therapeutics: A Multidisciplinary Viewpoint

**DOI:** 10.3390/cells10113097

**Published:** 2021-11-09

**Authors:** Mohammad Yahya Momin, Ravinder Reddy Gaddam, Madeline Kravitz, Anisha Gupta, Ajit Vikram

**Affiliations:** 1Mylan Laboratories Ltd. R&D Centre, Bangalore 560105, Karnataka, India; mominzy@gmail.com; 2Division of Cardiovascular Medicine, Department of Internal Medicine, Carver College of Medicine, The University of Iowa, Iowa City, IA 52242, USA; ravinderreddy-gaddam@uiowa.edu; 3Department of Pharmaceutical Sciences, University of Saint Joseph, West Hartford, CT 06117, USA; mkravitz@usj.edu (M.K.); agupta@usj.edu (A.G.)

**Keywords:** peptide nucleic acids, miR delivery, miRNA, plant-derived miRs

## Abstract

microRNAs (miRs) are emerging as attractive therapeutic targets because of their small size, specific targetability, and critical role in disease pathogenesis. However, <20 miR targeting molecules have entered clinical trials, and none progressed to phase III. The difficulties in miR target identification, the moderate efficacy of miR inhibitors, cell type-specific delivery, and adverse outcomes have impeded the development of miR therapeutics. These hurdles are rooted in the functional complexity of miR’s role in disease and sequence complementarity-dependent/-independent effects in nontarget tissues. The advances in understanding miR’s role in disease, the development of efficient miR inhibitors, and innovative delivery approaches have helped resolve some of these hurdles. In this review, we provide a multidisciplinary viewpoint on the challenges and opportunities in the development of miR therapeutics.

## 1. Introduction

miRs are small noncoding RNAs that bind to the 3′ untranslated region (3′UTR) of target genes and regulate their expression. The discovery of lin-4 and let-7 in the 1990s demonstrated that small noncoding RNAs have a general role in gene expression regulation [1,2]. miRs are found in eukaryotes, with some exceptions (e.g., Ctenophora and red algae) [3,4], and an increase in the number of miRs during animal evolution supports their involvement in the development of animal complexity. Humans have over 2000 miRs, and they are known to regulate the expression of thousands of genes [5,6]. A change in the miR expression during disease conditions, the ability of miRs to target hundreds of genes, including those that encode for traditionally undruggable proteins, and their small size have established them as an attractive therapeutic target. miR’s ability to bind to the target gene can be modified by multiple approaches, including the use of miR mimics, miR inhibitors, miR-mRNA-specific target site blockers, and decoy sponges. Several such molecules have shown efficacy in preclinical (e.g., miR-29, miR-451, miR-15, and miR-195) and in the early stages of clinical trials (e.g., miR-17, miR-34a, and miR-122) [7]. Overall, the moderate efficacy, associated adverse effects, and poor delivery of these molecules to the target tissues have impeded the development of miR therapeutics. This review employs a multidisciplinary approach to understand the challenges, discuss the novel delivery approaches, and identify the novel opportunities in developing miR therapeutics.

## 2. The MiR Biogenesis and Mechanism of Action

The miR encoding genes are located in the coding and noncoding regions of the genome. The miRs located in the introns of protein-coding genes are often co-expressed with the host genes but can also be expressed independently. For example, miR-204 is located in the *TRPM3* gene, and multiple studies have demonstrated their co-expression [8,9,10,11,12]. However, the von Hippel-Lindau tumor-suppressor gene (VHL) deletion-induced change in miR-204 expression correlated only with the short transcripts from the *TRPM3* gene but not with the large transcript that encodes for the protein [13], which suggests a host gene-independent transcription of intragenic miR. In contrast to intronic miRs, the transcription of intergenic miRs relies on their own promoter, and the molecular mechanisms remain largely elusive. Recently, Batista et al. reported that the expression of intergenic miR-200c/141 is linked with the proximal upstream gene PTPN6 (SHP1) during oxidative stress [14].

The miR-encoding genes are transcribed to primary miRs (pri-miRs), which are typically spliced, capped, and polyadenylated [15]. The Drosha-DGCR8 complex processes the pri-miRs and generates precursor miRs (pre-miRs) that are about 70 nucleotides long, adopt a double-stranded hairpin conformation, and are transported out of the nucleus by exportin-5 in a Ran-GTPase-dependent manner [16]. In the cytosol, the ribonuclease Dicer cleaves the pre-miRs to generate two mature miRs [17]. The miRs bind to the argonaute (AGO) protein, forming the miR-induced silencing complex (miRISC). Often, one strand is loaded onto miRISC while the other one is degraded. The miR strand sorting depends on multiple factors, such as tissue types, development stages, disease, and the abundance of target genes [18,19,20]. However, whether the absence of a complementary sequence during miR deletion studies affects the precursor hairpin stability or arm sorting remains unknown. In mammals, AGO proteins include AGO1-4, and all bind to and contribute to miR activity. However, only AGO2 has mRNA-cleaving catalytic activity and is therefore known as a slicer [21]. miR guides the miRISC to its target mRNA by sequence-specific complementarity. Upon binding to the target gene, miRISC destabilizes the gene via the RNase activity of AGO2 and inhibits the translation by preventing ribosomal complex formation. The sequence complementarity between the miR (seed region, nucleotides 2–7 of the miRs) and target mRNA determines the AGO2-dependent mRNA degradation or miRISC-mediated translation inhibition [22]. An extensive sequence complementarily triggers the endonuclease activity of AGO2 [22,23]. Some miRs are differently processed, such as, after Drosha processing, pre-miR-451 binds with AGO2 instead of Dicer, which cleaves one strand [24,25]. Similarly, the biogenesis of some miRs (mirtrons; miRs located in the short introns of host genes) is Drosha-independent, as the spliced-out introns serve as pre-miRs and are directly processed by the Dicer [26,27]. Kim et al. reported that miR-3615 and 7706 do not require Drosha processing [28].

miRs stoichiometrically load onto the miRISC, and the quantitative analysis shows that the total miR is over 13-fold higher than the AGO proteins in the HeLa cells. Further, the miR:AGO complex is seven-fold higher than the free AGO proteins [23]. This suggests that a significant proportion of miRs does not bind to the miRISC, and the biological role of such miRs remains unknown. miR-328 binds to hnRNPE2, leading to increased translation of the C/EBP protein in an example of the RISC-independent effects of miR, but this is still a rare mechanism, as miR-328 primarily acts via the canonical mechanism [29]. Under specific conditions (e.g., serum starvation and in quiescence), the miRs also initiate gene translation, and that involves interactions between AGO2 and FXR1 instead of AGO2 and GW182 [30,31]. Most of the mechanistic studies investigating the miR effects on gene expression are based on proliferating cell lines where miR is expected to repress the target gene expression. However, many cells maintain the quiescent stage in healthy humans, and studies have shown the critical role of miRs in maintaining quiescence. For example, the absence of miR-489, leading to increased levels of the oncogene Dek, results in the exit of muscle stem cells from quiescence [32]. Katz et al. showed that miR-9 maintains quiescence in neuronal stem cells by potentiating Notch signaling, which involves a nuclear mechanism [33]. Recently, miR-218 has been shown to regulate endothelial cell quiescence via MYC repression. Truesdell et al. reported that the miR-mediated upregulation of target mRNAs involves nuclear entry of the miR [31]. Call et al. studied the miR-mRNA interactions in activated and quiescent human hepatic satellite cells and reported a change in the miR profile and a negative correlation between the miRs and target genes [34]. However, whether miR-dependent gene upregulation is the outcome of cell physiology adjustments or is required for cells to enter quiescence remains unknown. The binding of miR to the 5′UTR of mRNAs encoding ribosomal proteins also activates their translation [35]. The determination of the target genes of miRs is vital to determine their biological role, leading to their development as miR therapeutics or biomarkers. The target identification is either based on bioinformatic target prediction or experimental assays (e.g., expression analysis and luciferase assay) [36]. These findings show that miRs target 100s to 1000s of genes and primarily inhibit translation in a sequence-specific manner but can also initiate translation (Figure 1a).

## 3. MiRs: Therapeutic Target

miRs are increasingly recognized as a regulator of disease pathogenesis and represent an important molecular target for gene therapy. miR therapeutics aims to improve health conditions by targeting miRs. miR modulators present a unique opportunity to regulate the expression of proteins that cannot be targeted by the traditional drug molecules (termed “undruggable”). For example, Dicerna Pharmaceuticals, Inc. reported silencing oncogene protein myc using DICER substrate small interfering RNA (DsiRNA) reduced tumor volume in multiple mouse tumor models [37]. Here, we summarized the general approaches for miR modulation and delivery.

### 3.1. MiR Modulation

Multiple approaches are commonly used to change the miR availability for miRISC loading and their effects on the target genes (Figure 1b).

#### 3.1.1. Deletion of Processing Enzymes

The deletion of enzymes that control the processing of miR generation (e.g., Drosha and Dicer) will affect the global levels of miRs. The cardiomyocyte-specific deletion of Dicer leads to dilated cardiomyopathy, cardiac remodeling, and heart failure [38,39]. Similarly, the conditional deletion of Dicer in the neocortex led to cortex hypotrophy and death of the pups soon after weaning [40]. Besides, a mouse model with deletion of the DGCR8 gene, which encodes for the microprocessor complex subunit, changed the miR biogenesis and contributed to the behavioral and cognitive impairments [41]. The deletion of either Drosha or Dicer in mouse skin epithelial cells also prevented the development of hair follicles [42]. As both enzymes have distinct functions, a similarity of the phenotypic outcomes suggested the involvement of miR biogenesis [42]. As 1000s of miRs are expressed, and each has a complex role in gene regulation, the deletion of the miR-processing enzyme is unlikely to be utilized for miR therapeutics.

#### 3.1.2. MiR Mimics and MiR Precursors

miR mimics have the identical sequences as endogenous mature miRs, and they are used to increase the levels of miRs [12,43]. As mimics do not introduce any new sequences, the synthesized miR does not have any vector-based toxicity. A vector-based technique where the miR precursor sequence is inserted downstream of the RNA polymerase-driven promoters is used to achieve the long-term upregulation of miRs [44,45]. miR mimics increase the levels of miR and can be developed as miR therapeutics (e.g., MRG-201 designed to mimic the activity of miR-29 [46] and MRX34 designed to mimic the activity of miR-34a [47]). Similarly, the miR precursors also increase the levels of mature miRs but are traditionally considered transitory intermediates for mature miRs. However, recent studies have shown the regulatory role of precursor miRs in target recognition [48,49]. Jafari et al. demonstrated a difference in the anticancer effects of miR-34a and pre-mir-34a and attributed these effects to the stable expression of pre-mir-34a and their regulatory roles [50]. Interestingly, fluoroquinolone antibiotic enoxacin promotes the processing of miRs and induces the expression of specific miRs (e.g., miR-125a, miR-139, miR-199b, and miR-23b) [51]. Though this is still a nonspecific approach for increasing the miR expression, Young et al. screened the libraries of small organic molecules and identified a molecule that explicitly upregulates miR-122 (e.g., 2-(2-(Dimethylamino)ethyl)-5-amino-1*H*-benzo[de]isoquinoline-1,3(2*H*)-dione) [52].

#### 3.1.3. MiR Inhibitors and MiR Sponges

In the past couple of decades, antisense oligonucleotides (ASOs) technology has seen unprecedented development and offers multiple classes of molecules that serve as miR inhibitors. Recently, we systematically discussed the chemical modifications, mechanism of action, and optimized delivery strategies of several different classes of ASOs [53]. Briefly, the locked nucleic acid (LNA) and phosphorothioate-based ASOs are enzymatically stable; upon binding to the target miR, they induce RNase H1 cleavage and have a high affinity to the target miR. Owing to the negative charge on the backbone of LNA and phosphorothioate-based ASOs, they nonspecifically interact with the proteins, leading to decreased urinary clearance and an increase in the half-life (typically ranges in weeks). The advantage of binding with a protein is that these molecules require a lower dose to achieve the desired miR inhibition, which entails accumulation in nontarget tissues and adverse outcomes. Peptide-nucleic acid (PNA)-based ASOs and 5′ methylcytosine-nucleobase-modified ASOs have a higher binding affinity to target miRs, act by steric hindrance, and are not associated with immune stimulation. PNAs have poor water solubility and rapid urinary clearance, and for that reason, they did not progress as the chemistry of choice for miR inhibition. The various chemical modifications to PNAs (e.g., cationic PNAs, α and γ guanidinium PNAs, and lysine PNAs) are employed to improve PNA’s water solubility and affinity to target miR [54,55,56,57]. Young et al. screened the libraries of small organic molecules and identified that Acetyl-*N*-(naphthalene-3-yl) benzamide, *N*-Hexyl-1,2,3,4-tetrahydroquinoline-6-sulfonamide specifically inhibits miR-122 [52]. In contrast to inhibitors, the sponge transgene is often delivered through a virus, and the transcribed mRNA has multiple binding sites for the target miR. This method of miR inhibition offers some advantages over the miR knockout or the use of a miR inhibitor for loss-of-function studies. For example, the miR knockout approach becomes challenging when closely related miRs are encoded from different loci, or miR precursors are transcribed in clusters. The proximity of the miRs within a cluster makes it difficult to delete one miR without affecting others. As the sponges interact with the mature miR, their effectiveness is unaffected by the multiple miRs with closely related seed sequences or the clustering of miR precursors [58]. Many cells are difficult to transfect by ASOs but can be easily infected by viruses carrying a transgene for the miR sponge. Besides, for the continuous inhibition of miRs, ASOs need to be transfected repeatedly while the continuously transcribed miR sponge offers extended inhibition of the miRs.

#### 3.1.4. Target Site Blockers (TSBs)

These molecules are single-stranded RNA that specifically block the binding site of the miR on the 3′UTR of the target gene. TSBs do not affect the binding of miR to other target genes. Therefore, these molecules help verify whether the miR interaction with a specific gene contributes to the observed effects. For example, TSBs that stop the interaction between miR-145-5p and the CFTR gene increase the gene expression and anion channel activity, suggesting that the interaction of miR-145-5p with the CFTR gene is responsible for this effect [59]. Similarly, the use of TSB confirms the interaction of miR-155 with PAD4 for the neutrophil extracellular trap during inflammatory diseases [60].

### 3.2. Delivery of MiR Modulators

The selection of the miR delivery system depends on the nature of miR therapeutics (enzyme susceptibility), the expression pattern of the target miR (for miR inhibitors) or gene, the intended site of delivery, and the adverse effect tolerability. Delivery to the central nervous system possesses the additional challenge of crossing the blood-brain barrier. For example, despite the high delivery efficiency of viral vectors, activation of the host immune response is a concern; lipid-based particles deliver primarily to the liver and the reticuloendothelial system [61], while large-sized lipid-based particles can be advantageous, as they escape renal filtration, allowing a higher payload [62]. Here, we summarized the use of viruses, inorganic materials, 3D scaffolds, polymers, and lipids to deliver miRs and miR inhibitors (Figure 2).

#### 3.2.1. Viral Vector-Based Delivery System

Viruses are a submicron biological system that actively infects animal and plant cells and intracellularly releases their genomes [63,64]. The infecting characteristics of the virus are exploited to deliver the genes of interest [65]. The intravenous injection of recombinant adeno-associated virus-encoding miR-122 inhibitor depleted miR-122 expression and increased the expression of its target genes, and led to a >30% decline in the serum cholesterol level in mice [66]. Adeno-associated virus serotype 9 (AAV9) has been used to deliver miR-196a to treat spinal bulbar muscular atrophy in mice [67]. The viral delivery system Glybera^®^ (alipogene tiparvovec) is commercially available for severe episodes of pancreatic inflammation associated with lipoprotein lipase deficiency despite a low-fat diet intake [68]. Despite the potential of viral vectors as a promising and practicing gene delivery vehicle, the induction of the host immune response is a major concern. Mutagenesis and oncogenesis make viral vector-based therapy unsafe, especially while using a virus that employs transgene integration with the genome [69].

#### 3.2.2. Inorganic Material-Based Delivery System

Gold nanoparticles, mesoporous silica, graphene oxide, and ferric-oxide-based nanoparticles have been utilized to deliver miR inhibitors. The gold nanoparticle-based miR-155 inhibitor delivery to macrophages improved cardiac function in ovariectomized diabetic mice [70,71]. Mesoporous silica nanoparticles efficiently delivered the miR-155 inhibitor to colorectal cancer cells in vitro and in vivo [72]. Recently, hyaluronic acid-conjugated graphene-oxide loaded with the Cy3-tagged miR-21 inhibitor has been shown to target CD44-positive MBA-MB231 cells [73]. The systemic administration of ferric-oxide-based nanoparticles carrying miR-100 enhanced the antitumor effects of docetaxel in FGFR3-mediated patient-derived xenografts [74].

#### 3.2.3. D Scaffold-Based Delivery System

Some recent studies have demonstrated the use of 3D scaffolds to deliver miRs. Hydrogel is a polymeric network with hydrophilic properties, and PEGylation hydrogel has been shown to release miR-20a and promote the differentiation of mesenchymal stem cells into osteoblasts [75]. A self-assembled RNA triple-helix hydrogel consisting of miR-205 and anti-miR-221 led to 90% shrinkage in the tumor volume two weeks post-gel implantation in a triple-negative breast cancer mouse model [76]. Zhang et al. reported that a hyperbranched polymer could self-assemble into nanoscale complexes, deliver miR-26, and reduce calvarial bone defects in an osteoporotic mouse model [77].

#### 3.2.4. Polymeric- and Dendrimer-Based Delivery Systems

Polyethylenimines (PEIs), poly (lactic-co-glycolide) (PLGA), and chitosan have been used for the delivery of miR mimics and inhibitors. Poly-l-lysine modified PEI loaded with the miR-21 inhibitor miR-21 levels in MCF-7 cells [78]. Cai et al. showed that PLGA-based nanoparticles tagged with anti-vascular endothelial growth factor antibodies are efficiently taken up by hepatic carcinoma cells and deliver miR-99a [79]. Chitosan is a natural and biocompatible polymer that contains d-glucosamine and *N*-acetyl-d-glucosamine [80]. As macrophages express high levels of galactose/*N*-acetyl-galactosamine-specific lectin, glycosylated low molecular weight chitosan (G-LMWC) targets macrophages [81]. Huang et al. reported that G-LMWC combined with miR-16 precursors increased the macrophage miR-16 levels, reduced the TNF-α and IL-12p40 expression, and rescued the colic symptoms induced by 2,4,6-trinitrobenzene sulfonic acid in mice [82].

Dendrimers are synthetic polymers with a structure of repeated branching chains that form three-dimensional spherical macromolecules. PAMAM, a synthetic polymer with a dendrimeric structure, is more biocompatible and biodegradable than PEI. Recently, it has been demonstrated that the polyethylene glycol-modified graphene-oxide and PAMAM dendrimer conjugate efficiently delivers miR-21 to non-small-cell lung cancer cells and targets lung adenocarcinoma tumors in vivo [83].

#### 3.2.5. Lipid-Based Delivery Systems

Lipids have been widely used as excipients in the delivery carriers due to the ease of chemical modifications, allowing molecules’ attachment to target-specific tissues. Lipid-based delivery systems include liposomes, lipid nanoparticles, and solid lipid nanoparticles (SLNs) (Figure 3). They are used for nucleic acid delivery. The intracellular delivery of negatively charged miRs/miR inhibitors could be challenging, as the cell membrane is also negatively charged, but cationic lipids can carry them as cargo to deliver to the target site [84]. Lipofectamine^®^ is a lipid that consists of a 3:1 mixture of DOSPA (2,3-dioleoyloxy-*N*-[2(sperminecarboxamido)ethyl]-*N*,*N*-dimethyl-1-propaniminium trifluoroacetate) and DOPE (1,2-Dioleoyl-sn-glycero-3-phosphoethanolamine) and is capable of forming liposome vesicles and acting as a transfecting agent for the intracellular delivery of RNAs and plasmid DNA by lipofection [85].

Liposomes consist of lipids with a hydrophilic head and hydrophobic tail that orient as concentric bilayers forming an aqueous core [80,86]. If the lipid bilayer repeats concentrically, it forms multilayered vesicle liposomes, and if a single vesicular structure repeats in a nonconcentric way, it forms multivesicular liposomes. Liposomes use a mixture of anionic, cationic, and neutral lipids. Cationic lipids are beneficial for miR delivery, as they conjugate with negatively charged miRs for easy loading into the aqueous portion of liposomes and interact with the target cellular membrane for miR delivery [87]. However, the cationic nature of liposomes causes cellular toxicity through interaction with the negatively charged cell membrane and forms aggregates with serum proteins that also bear a negative charge [88,89]. To overcome this problem, neutral lipids are mixed with cationic lipids to confer stability and reduce positive charge-mediated unwanted consequences [90]. Yan et al. showed that miR-203-loaded liposomes silence slug expression in vitro and in vivo, inhibit the TGF-β1/Smad pathway in triple-negative breast cancer cells, and improve the anticancer effects of chemotherapy [91]. miR-7-loaded cationic liposomes inhibit epidermal growth factor receptor (EGFR) and suppress the spread of ovarian cancer [92]. Moreover, Zhang et al. reported that transferrin-targeted miR-221 inhibitor-loaded liposomes show a 15-fold higher delivery to HepG2 cells than nontargeted liposomes and increase the expression levels of miR-221 targets, suggesting efficient delivery [93].

Lipid nanoparticles (LNs) are liposome-like nanosized vesicles that consist of pH-sensitive lipids that are positively charged in acidic pH and neutral at physiological pH. In contrast to liposomes, LNs have a single lipid layer with a considerable volume for encapsulating cargo, and multiple micelles are formed within the core [94]. LNs form complexes with nucleic acids (e.g., RNA) in a cationic state of the lipid (at a low pH) and avoid an endosomal escape as they acquire a neutral charge in the body [94]. LNs were recently used for COVID-19 mRNA vaccine development by BioNTech and Pfizer [95]. Patel et al. tested different types of lipid nanoparticles for mRNA delivery into the posterior of the eye to prevent blindness [96]. The small sizes of the particles and pH-based endosomal escape make them particularly attractive delivery vehicles [97]. However, LNs have a limited loading capacity and high uptake by the liver and spleen [96,98].

Solid lipid nanoparticles (SLNs) are produced substantially from the solid lipidic materials comprising the lipid matrix. In contrast to liposomes and LNs, the cargo in SLNs is embedded into the matrix instead of being encapsulated. Surface active agents confer stability and ligand properties during the formulation [99]. Due to the lower toxicity profile and high biocompatibility, SLNs have been explored to deliver therapeutics and received approval for pharmaceutical applications in humans [100,101]. The synthesis of SLNs does not require an organic solvent. They have a lipidic matrix within which drug/miR can be entrapped, leading to long-term stability to the formulation [102,103,104]. A combined delivery of paclitaxel and miR-200c using cationic SLNs enhanced the susceptibility of breast cancer stem cells to paclitaxel by inhibiting class III β tubulin expression [105]. Further, SLNs were used for inhibiting miR-21, and the results showed the suppression of miR-21 and inhibition of lung cancer growth [106].

#### 3.2.6. Cell-Based Delivery Systems

The structure of liposomes represents a synthetic cell whose development is time-consuming and expensive, while naturally occurring cells are readily available as a substitute for drug delivery applications. The physiological roles of cells and spatiotemporal distribution of cell-based delivery systems that decrease the toxicity and increase the therapeutic efficacy make them suitable delivery systems [107,108,109]. We describe some cell-based miR delivery systems.

Red blood cells (RBCs) are actively involved in oxygen and carbon dioxide transportation, and their unstoppable journey confers transportability advantages. RBCs are known to have a small amount of miRs and mRNAs [110]. In contrast to free nanoparticles, RBC-associated nanoparticles have been shown to deliver ten times more drugs to the target site [111]. The half-life of RBC is 10–15 days in comparison to PEG liposomes (3–6 h), filomicelles (1–3 days), polymeric micelles (0.1–6 h), proteins and conjugates (10 min–6 h), and polymersomes (10–12 h) [112]. Due to an extended half-life compared to other artificial career systems, RBCs remain longer in the blood circulation, reaching all organs and tissues throughout the body. The delivery of therapeutics using RBCs could be achieved by encapsulating drug-loaded nanoparticles within or by attaching them to the surfaces of erythrocytes [112]. Usman et al. demonstrated that RBC-derived extracellular vesicles deliver the miR-125b inhibitor and reduce cancer progression [113].

T cells are omnipresent in the body, though most are present in lymphoid tissues [114]. These cells recognize, attack, and eradicate cancer cells [115] and, thus, can act as a naturally occurring delivery vehicle for anticancer modalities. As these cells are nucleated, the abundance of nucleic acids might interact and dilute the efficacy of the miR/miR inhibitor. However, the miR/miR inhibitor encapsulation within a T cell could protect it from blood ribonuclease, minimize its interaction with the cellular DNA/RNA, and selectively deliver miRs at the target site. Recently, Parayath et al. reported that injectable nanocarriers providing in vitro transcribed chimeric antigen receptor mRNAs transiently reprogram circulating T cells to recognize disease-relevant antigens [116,117]. They demonstrate that the infusion of these nanocarriers induces host T cells to express tumor-specific chimeric antigen receptors to cause disease regression as efficiently as achieved by the infusion of ex vivo engineered lymphocytes [116,117].

Dendritic cells (also known as antigen-presenting cells) play a crucial role in regulating immune responses [118,119]. Firdessa-Fite et al. compared a mRNA-based dendritic cell delivery system against lipid nanoparticles targeting CD4^+^ and CD8^+^ T cells. The injection of both delivery systems via different routes resulted in mRNA delivery to the respective sites, instigating T-cell responses [120]. These cells were explored for the delivery of miRs. The intravenous administration of a Cy5-tagged miR let-7-loaded extracellular vesicle conjugated with aptamer AS1411 decreased the tumor growth in vivo [121]. Given the efficient delivery of oligonucleotides, dendritic cells hold potential for miR and miR inhibitor delivery.

Macrophages are monocyte-derived cells that actively phagocytose foreign cells, cancerous cells, dead cells, dying cells, and microbes. The tumor-targeting ability of macrophage-derived exosomes and their cell membranes can be harnessed as a tool for delivering miRs or miR inhibitors to tumors [122]. The macrophages take up liposome-entrapped RNA molecules, some of which remain undegraded and are released via microvesicles. Wayne et al. demonstrated that macrophages loaded with calcium integrin-binding protein-1 (CIB1) siRNA result in decreased tumor growth and decreased mRNA expression of CIB1 and KI67 in MDA-MB-468 human breast cancer cells [122]. Akao et al. demonstrated that the chemically modified miR-143 containing microvesicles was secreted from THP-1 macrophages following intravenous injection [123]. These results suggest the suitability of using macrophages-derived vesicles as a delivery system for miR inhibitors. Besides, as macrophages are heavily involved in the pathogenesis of multiple diseases, understanding how to use macrophages for drug delivery can benefit the treatment efficiency.

Natural killer cells migrate to the inflammation site and attack cancer cells; therefore, they can be used for the drug delivery of anticancer miR modulators. Neviani et al. reported that the natural killer-derived exosome miR-186 inhibits neuroblastoma growth and immune escape mechanisms [124]. Further, natural killer-derived exosomal vesicles inhibit the malignant transformation of pancreatic cancer, which requires the presence of miR-3607-3p [125].

## 4. Challenges in the Development of MiR Therapeutics

Less than twenty miR therapeutics have entered clinical trials, and none progressed to phase III trials. In contrast, over 60 siRNA drugs are in clinical trials, with two approvals [126]. In a direct comparison between miR therapeutics and siRNA therapeutics, a higher percentage of terminated/suspended clinical studies (50% vs. 35%) and none in phase III clinical trials (0% vs. 12%) suggest unexplainable hurdles in the development of miR therapeutics [126]. However, this could be because of the fewer miR therapeutic candidates in the pipeline. Exogenous siRNAs are designed explicitly against the target gene. On the other hand, endogenous miRs target multiple genes. Zhang et al. analyzed the targets of miRs and siRNAs that entered the clinical trials and found that miR targets ranged between 30 and 250 while that of siRNAs ranged between 1 and 3 [126]. The siRNAs exhibit 100% sequence complementarity with the target gene, whereas the sequence complementarity of endogenous miRs range between 20 and 90% [126]. miRs target tens to hundreds of genes and inhibit gene expression by decreasing the stability or inhibiting the translation. Still, in specific conditions (e.g., starvation), they upregulate the expression of genes. The disease pathway analysis of a miR-34a mimic predicted the deregulation of two immune pathways, and it was discontinued in phase I clinical trial due to immune-related severe adverse effects [126]. Nonetheless, the disease pathway analysis of miR-122 showed only a small number of genes linked with jaundice, and it did not appear in the phase I subjects but appeared in the few cases in phase II [126]. Therefore, the multiple targets of miRs are a crucial challenge in the development of miR therapeutics. Although this is the inherent characteristic of miRs, this limitation can be debilitated by targeted delivery and chemical modifications to the molecules that improve their binding characteristics to the target genes. Further, the use of miRs derived from the plants that affect a gene that belongs to the pathogen (e.g., aly-miR396a-5p) and does not target genes in the host could be utilized to develop miR therapeutics.

The development of miR therapeutics would require the identification of biologically significant miR in a disease condition, efficient delivery of miR modulators to the target cell/tissue, regulation of target(s) expression, and minimal nonspecific interactions. The biological complexity and physiochemical nature of the miR/miR inhibitor pose challenges in developing miR therapeutics, as described below (Figure 4).

### 4.1. The Negative Charge of the MiR Inhibitors

Most DNA/RNA-based miR modulators are negatively charged, promoting their nonspecific binding to the blood proteins and decreasing urinary clearance [127]. As an example, a phosphorothioate-modified single-stranded oligonucleotide, which is negatively charged, is quickly transmitted from the blood to tissues (minutes to hours) and exhibits a half-life of 2–4 weeks. In contrast, oligonucleotides that lack a charge (e.g., PNAs and unmodified siRNA) weakly bound to plasma proteins exhibit a rapid clearance either due to metabolism in the blood or excretion via urine, leading to a lower tissue uptake [127,128]. Therefore, the nonspecific accumulation of ASOs in tissues could lead to sequence complementarity-dependent effects in nontarget tissues. The common concerns with the synthetic nucleic acid analogs include (a) a moderate efficacy and (b) adverse effects, such as prolonging the activated partial thromboplastin time, activating the complement cascade, and accumulating nonspecifically in tissues. The activated partial thromboplastin time depends on the plasma concentration of the nucleic acid analogs and is not considered clinically significant [129,130,131]. However, the accumulation of nucleic acid analogs in tissues contributes to adverse outcomes that depend on the negative charge of these analogs and leads to nonspecific binding to proteins [132]. The negative charge on miR modulators negatively affects the cellular entry of these molecules via passive diffusion as the cell membrane is also negatively charged.

### 4.2. Functional Complexity and Off-Target Effects

A single miR targets hundreds to thousands of mRNAs, and a single mRNA is targeted by tens to hundreds of miRs, making identifying one specific miR that controls a disease pathogenesis challenging. Although in silico tools can predict the binding strength of a miR to the target mRNA, the biological effect depends on multiple factors, such as expression of miR, availability of AGO2, the abundance of the target gene of interest, and expression of the miR target genes. Besides, the expression of target genes in other cell types and tissues can be targeted by miR modulators, which often contribute to unwanted effects. Most of the DNA-based miR modulators have a negative charge that leads to their binding with macromolecules, leading to nonspecific tissue accumulation and prolonged tissue half-lives (2–4 weeks). Therefore, identifying the miR that plays a therapeutically significant role in disease pathogenesis is a challenge.

### 4.3. Stability of MiR Modulators in the Blood and Endosomal Escape

miRs and unmodified miR inhibitors are prone to degradation by ribonuclease enzymes in a biological system [133]. Naked miRs with an unmodified 2′ OH in ribose are quickly degraded in the blood by the serum RNase [134]. Modifying 2′ OH (e.g., locked nucleic acid, phosphorothioate, *O*-(2-Methoxyethyl), *O*-Methyl, and Fluro) can improve the stability and affinity of the target genes [134,135]. Ribonuclease enzymes that degrade miRs invade the duplex via the sense strand’s less stable terminus (3′ terminus) [136,137]. Chemical modifications have been developed to prevent this [136,137]. Further, chemical modifications are not well-tolerated at the 5′ end of the antisense strand, highlighting how crucial molecular asymmetry plays in RNAi activity [138]. This evidence suggests that phosphorothioate modifications at the sense strand prevent degradation and improve serum stability. The intracellular trafficking of miRs begins in the early endosome, which later merges with late endosomes and lysosomes with acidic environments where nucleases degrade them. For the miRs inhibitors to engage with miRs and the miR mimic to engage with miRISC, they will need to escape the endosome, which constitutes a challenge. Strategies such as pH-sensitive lipoplexes/polyplexes and photosensitive molecules can promote an endosomal escape [139,140].

### 4.4. Chemical Modifications and Cellular Entry of MiR Inhibitors

A vital issue in miRs therapeutics is the sequence-independent and complement-mediated toxicity owed to the chemical modifications (e.g., phosphodiester linkages, ribose backbone, and 2′-*O*-(2-methoxyethyl)) made in an attempt to mitigate the nuclease-mediated degradation of miR modulators [136,141]. The inhibition of blood coagulation, complement cascade activation, immune cell activation, and reduced peripheral white blood cell counts are often observed as signs of toxicity arising from such chemical modifications [138]. Phosphorothioate-based miR inhibitors exhibit poor uptake in the skeletal muscle and brain due to their hydrophilic and negatively charged nature and inability to cross tight junctions, such as the blood–brain barrier [136,138]. However, these molecules are stable; exhibit a long half-life; and show excellent penetration of some tissues (e.g., kidney, liver, spleen, lymph nodes, and bone marrow) [138]. The sizes of miR inhibitor-loaded nanoparticles determine their ability to penetrate through the pores of leaky vessels. The pH affects the formation of smaller and monodispersed nanoparticles, while the outer surface hydrophilicity of nanoparticles determines their urinary clearance [142,143]. Thus, in addition to optimizing the chemical nature of miR inhibitors, the optimization of chemical conditions during nanoparticle synthesis and modifications of the molecules on the outer surfaces of nanoparticles can affect the delivery of the miR inhibitors to the target tissues. While the various deliveries, as mentioned above, alongside cautious dosing, resolve this concern to some extent, further studies are warranted to generate biocompatible and efficient miR therapeutics.

## 5. Opportunities for the Development of MiR-Therapeutics

The development of more biocompatible and efficient miR inhibitors (e.g., γ-modified PNA-based miR inhibitors), the identification of plant-based miRs with biologically significant effects in animals by targeting the expression of a specific gene (e.g., aly-miR-396a-5p), and the microvesicle-dependent targeted delivery of miRs are emerging opportunities for the development of miR therapeutics (Figure 4).

### 5.1. Gamma-Modified PNA-Based MiR Inhibitors

Several synthetic nucleic acid analogs have been employed to target miRs. However, various challenges still need to be resolved for miR inhibitor-based therapeutic modalities to be safe and effective in patients. Peptide nucleic acids (PNAs) are neutral charge synthetic DNA/RNA mimics in which the phosphodiester backbone is substituted with an *N*-(2-aminoethyl) glycine backbone [144,145,146]. Due to poor water solubility, the regular PNAs did not progress as the chemistry of choice. However, a novel class of PNAs called gamma PNAs (γPNAs) are highly water-soluble and charge-neutral; they neither aggregate nor adhere to surfaces or other macromolecules in a nonspecific manner [147,148]. As individual strands, γPNAs adopt a right-handed helical motif—as confirmed by circular dichroism, nuclear magnetic resonance, and X-ray crystallography—and hybridize to DNA or RNA with an unusually high affinity and sequence specificity 54. Prior studies revealed that, on average, each γ modification stabilizes a PNA–RNA duplex by 5 °C [149]. γPNA is the only molecule known to invade duplex RNA/DNA composed of up to 100% CG content in a sequence-specific manner [53]. Another attractive γPNA design is tail-clamp modifications, which can be used to target miRNA-containing homopurine stretches [150].

Cheng et al. demonstrated that the γPNA-based miR-155 inhibitor attached to a pH-induced transmembrane structure produces a novel construct that targets the tumor and inhibits its growth in vivo [28]. In another study, the pre-radiation administration of γPNA-based antisense oligonucleotide against double-stranded break repair factor KU80 significantly increased the antitumor effects of radiation without showing any evidence of toxicity [151]. In two additional studies, γPNA-based miR inhibition showed efficacy against ischemia-reperfusion injury [152] and cancer [153] in vivo. Gupta et al. showed that γPNA delivered via polymeric nanoparticles antagonized the expression of miR-210 in a HeLa cancer model, resulting in an anticancer effect with no observed toxicities [154]. Our recent study has explored the potential of tail-clamp γPNA to improve the miR inhibition efficacy and decrease tumor progression in vivo [150].

γPNAs have been exploited in several biological and biomedical applications (e.g., electronic barcoding [155], nanotechnologies [156], gene editing [157,158,159,160], and gene targeting [151,161]). A comprehensive cytokine and metabolic analysis supported the in vivo biocompatibility of γPNAs [155]. Compared to DNA-based miR inhibitors, PNAs are rapidly eliminated via urinary clearance (blood half-life: hours vs. weeks), possibly because of their minimal interaction with the blood proteins [128,162]. Beavers et al. studied the blood pharmacokinetics of the intravenously injected PNA (non-gamma-modified)-based miR-122 inhibitor and found that the blood half-life was 0.9 min, which was increased to >30 min when loaded onto porous silicon nanoparticles [163]. They also measured the biodistribution of the Cy5-labeled PNA-based miR-122 inhibitor alone and loaded onto porous silicon nanoparticles 160 min after the intravenous administration, and found the presence of an inhibitor in multiple tissues (e.g., heart, lung, liver, and kidney) [163]. The lower retention may require a higher dosing frequency to achieve the desired miR inhibition and constitutes a limitation. However, it is outweighed by the exceptional affinity, specificity, and biocompatibility of γPNA-based miR inhibitors. Therefore, employing γPNA technology represents a novel approach to developing miR therapeutics.

### 5.2. Plant-Derived MiRs

Plant-derived exosomes contain miRs and can be taken up by the microbes residing in the gut or intestinal epithelial cells. The presence of the hyphen (-) symbol in animal miR distinguishes them from plant miR (e.g., miR-159 vs. miR159) [164]. Intestinal microbes take up plant-derived exosomes, and the lipid compositions of exosomes determine their selection by the microbes. For example, ginger exosome-like nanoparticles containing mdo-miR7267-3p are preferentially taken up by *Lactobacillaceae* [164]. mdo-miR7267-3p targets monooxygenase ycnE, which leads to increased levels of indole-3-carboxaldehyde and IL-22. This study showed that ginger exosome-like nanoparticle RNAs ameliorate mouse colitis via IL-22-dependent mechanisms [164]. Several studies have reported that plant-derived miRs are stable and cross the gastrointestinal tract (e.g., miR156a, miR168a, and miR159a/e) [165,166,167]. miR168a derived from *Moringa olifera* seeds was predicted to be a human analog of miR-579. Its transfection in the hepatocarcinoma cell line induced a significant decrease in the expression of SIRT1, an experimentally verified target of miR-579 [168]. Plant miR2911 shows better stability and gastrointestinal tract absorption than synthetic miR2911, which could be attributed to its association with plant proteins or plant-specific 2′-*O*-methylation [169]. Mlotshwa et al. tested the antitumor effects of an orally delivered cocktail of 2′-*O*-methyl-modified antitumor miRs (containing miR-34a, miR-143, and miR-145) and demonstrated a significant decrease in the tumor burden in mice [170]. The effect of ginger-derived miRs (aly-miR396a-5p and rlcv-miR-rL1-28-3p) depends on the specific inhibition of the spike protein and nonstructural protein 12 expressions [171]. In another study, the miR containing microvesicles isolated from *Moringa oleifera* (Drumstick) were bioinformatically predicted to target apoptosis-related human genes, and its exposure decreased the cancer cell viability and increased apoptosis [172]. From rice, multiple miRs (e.g., osa-miR156-5p and osa-miR164-5p) are predicted to target human genes [173]. Chin et al. reported the presence of plant miR159 in human serum and showed that its levels inversely correlated with breast cancer incidence and progression [165]. Further, they showed that miR159 targets TCF7, which is highly expressed in cancerous cells and effectively reduces the tumor volume in mice [165]. Thus, miR159 could potentially be used as a therapeutic agent against breast cancer. The precise uptake mechanism for plant-derived miR is not well-known, but miR-loaded exosome-like nanoparticle uptake by intestinal epithelial stem cells by micropinocytosis has been shown [174]. Further, stomach pit cells absorb dietary and orally administered miRs in a SIDT1-dependent manner [175]. Once internalized by the intestinal epithelial cells, they can be packaged into microvesicles and subsequently released into the bloodstream [176].

Although these reports greatly support the potential of plant-derived oral-administered miRs as cross-kingdom regulators, data reproducibility and technical errors are still a reason of concern. Dickinson et al. compared small RNA sequences in the serum and liver of mice fed a rice-based chow (up to 75% rice) with that of mice fed normal chow and found <10 reads per 10 million reads for the plant-derived miRs [177]. In another study, feeding mice with a diet supplemented with corn miRs for two weeks was not associated with a significant increase in their levels in the blood or tissues, suggesting the extensive degradation of miRs during the digestive process [178]. Further, the presence of diet-derived plant miRs at low levels and their presence in only a few animals (out of all that received a high corn diet) suggests that a leaky gut in mice caused by pathological conditions (e.g., cancer and inflammation) could contribute to the entry of orally administered miRs. The gastrointestinal uptake of miRs increased when mice were provided with aspirin or anti-CD3 antibodies, which increase gut permeability [179]. A comprehensive analysis of the sequencing dataset from human serum showed the presence of plant miRs at low levels, and no correlation between their levels in the tissues that were exposed to dietary miRs (liver) and that were not (cerebral spinal fluid) suggested the possibility of contamination [180]. However, the low reads for plant-derived miRs in the study by Dickinson et al. could be because of the sequencing bias of the approach for plant and animal miRs. As, unlike most of the previous high-throughput studies that detected the most abundant miRs (miR156 and miR168 of at least 10,000 reads per million), this study detected no miR-156 and shallow levels of miR168 (<200 per million) [181]. The plant-derived miRs have recently been actively discussed in the scientific community to understand their potential and loopholes as miR therapeutics [182,183]. We believe that plant-derived miRs are scalable, and pending validation by independent studies, they represent a unique opportunity to be developed as modulators of gene expression with therapeutic potential.

### 5.3. Cell-Targeted Delivery Using Microvesicles and Exosomes

Recent studies have shown the effectiveness of cell-derived microvesicles or exosomes in delivering oligonucleotides [121,123]. Several studies have shown that the transfer of miRs (e.g., miR-302b, miR-142-3p, and miR-223) by platelet-derived microvesicles affects the endothelial gene expression and inflammation of cells [184]. Greater than fifty percent of peripheral blood microvesicles are likely derived from non-nucleated platelets and have been shown to deliver miRs effectively. The experimental approach where platelet-derived microvesicles are loaded with the oligonucleotide of interest and modified to target a specific cell type, in our opinion, holds high promise as a delivery vehicle.

## 6. Conclusions

The causal role of miRs in multiple diseases, their small sizes, and specific targetability make them attractive drug targets. Despite these characteristics, the limited success with miR therapeutics could be attributed to too many targets of miRs, their ubiquitous expression, and sequence complementarity-independent effects in target/nontarget tissue leading to adverse outcomes. Even though the current failures with miR therapeutics are discouraging, miR’s ability to target multiple genes can be advantageous in managing complex disease conditions, as they also involve the deregulation of multiple signaling pathways. A parallel analogy is the use of natural products that regulate multiple targets to offer therapeutic benefits in contrast to modern medicine that acts on a specific target to mediate the desired effects [185,186]. The advances in understanding miR’s roles in disease pathogenesis, the development of nuclease-resistant γ-modified PNA-based anti-miRs and plant-derived miRs are novel approaches that help resolve or weaken some of the challenges and support the development of miR therapeutics.

## Figures and Tables

**Figure 1 cells-10-03097-f001:**
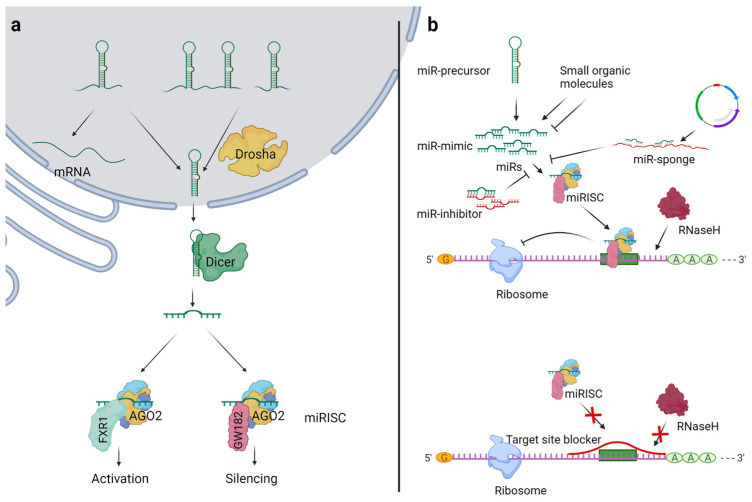
Overview of miR biogenesis and general strategies for miR modulation. (**a**) Drosha processes the primary miRs (pri-miRs) to precursor miRs (pre-miRs) in the nucleus, which are then transported to the cytosol by exportin 5. In the cytosol, the Dicer cleaves the pre-miRs to mature miRs. The miRs bind to argonaute-2 (AGO2), which interacts with GW182 to inhibit the target gene’s translation in a sequence-specific manner. In specific conditions such as starvation, AGO2 interacts with FXR1 and initiates the gene translation in a sequence-specific manner. (**b**) The use of miR mimics or miR precursors increases the levels of mature miRs. The miR inhibitor or miR sponge sequesters miRs and decreases their availability for miRISC loading. The target site blockers block the binding site of miR on the specific target mRNA, leading to mRNA-specific inhibition of the miR effects.

**Figure 2 cells-10-03097-f002:**
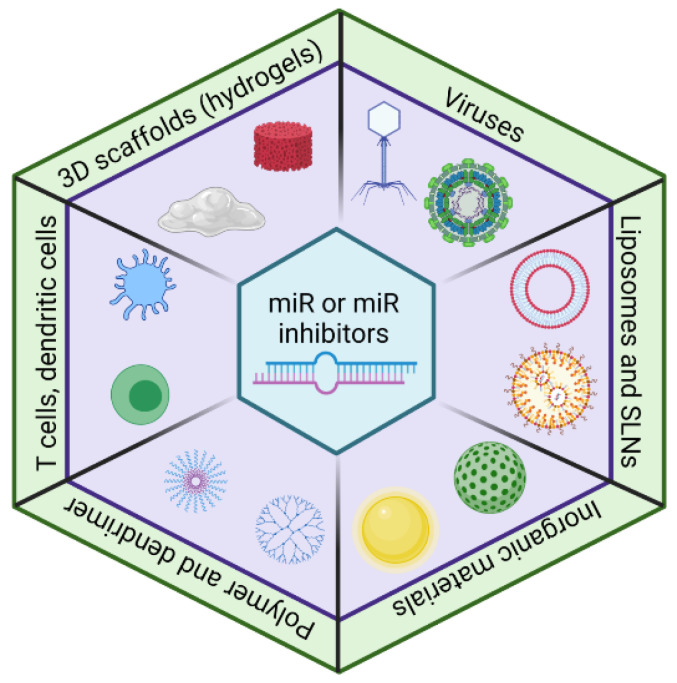
The delivery approaches for miRs and miR inhibitors. The viral vectors have high efficiency, but immunogenicity is a concern. A lipid-based delivery system (e.g., liposomes and solid lipid nanoparticles (SLNs)) is widely used for miR and miR inhibitor deliveries. Recently, inorganic material-based, polymer-based, cell-based, and 3D scaffold-based approaches have emerged as a delivery system for miRs and miR inhibitors.

**Figure 3 cells-10-03097-f003:**
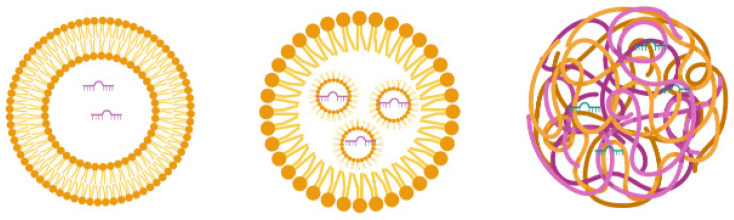
The general structure of the liposome, lipid nanoparticles, and solid lipid nanoparticles.

**Figure 4 cells-10-03097-f004:**
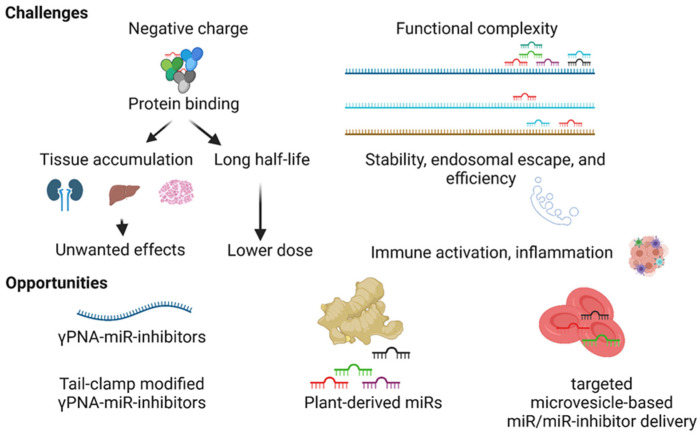
Challenges and opportunities in the development of miR therapeutics. The negatively charged miR inhibitors nonspecifically bind to blood proteins, promoting the half-lives and nonspecific tissue accumulation of molecules leading to unwanted effects. Hundreds of miRs target a single mRNA, and in turn, a single miR targets hundreds of genes, making target miR selection challenging. The stability, endosomal escape, and specificity of the miR inhibitor and adverse effects such as immune activation impede the development of miR therapeutics. Next-generation γ-modified PNA-based miR inhibitors, tail-clamp modification to the structures of miR inhibitors, plant-derived miRs, and targeted microvesicle-based delivery approaches present opportunities to resolve some of the challenges in the development of miR therapeutics.

## Data Availability

Not applicable.

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
