# Peer review of "The Challenges and Opportunities in the Development of MicroRNA Therapeutics: A Multidisciplinary Viewpoint"

_cells, 2021, doi:10.3390/cells10113097_

Round 1

Reviewer 1 Report

The interest for the role of microRNA as therapeutics and with innovative delivery approach in clinical practic has been growing in recent published literature. This review paper does bring large novelty to the subject, the described role of procedure that can detect the microRNAs in multidisciplinary approach to understand the challenges, discussing the novel delivery approaches, and identifing the novel opportunities in developing miR-theraapeutics. However, the manuscript needs a revision.

Comments and questions:

I would suggest to explian better the pharagraph “Delivery of miR modulators” This part is not well integrated with the rest of the text.

Sometimes the manuscript appears not well homogeneous in the various parts but unrelated to each other.

References do not seem up to date to the most recent literature on the subject.

Author Response

Comment: The interest in microRNA's role as therapeutics and with an innovative delivery approach in clinical practice has grown in recent published literature. This review paper brings a large novelty to the subject, the described role of procedure that can detect the microRNAs in a multidisciplinary approach to understand the challenges, discuss the novel delivery approaches, and identify the novel opportunities in developing miR-therapeutics. However, the manuscript needs revision.

Response: Thank you for finding the topic of this review novel.

Comment: I would suggest explaining better the paragraph "Delivery of miR modulators" This part is not well integrated with the rest of the text. Sometimes the manuscript appears not well homogeneous in the various parts but unrelated to each other.

Response: We have revised the "Delivery of miR modulators" to better integrate with the rest of the manuscript. Please see this section in the revised manuscript.  We made changes in the entire manuscript to make it more homogeneous. Please see the changes in the revised manuscript.

Comment: References do not seem up to date to the most recent literature on the subject.

Response: References have been updated.

Reviewer 2 Report

microRNAs (miRNAs) play an important role in the regulation of gene expression and disease pathogenesis. miRNA-therapeutics are the potential to be developed. However, less than 20 miRNAs targeting molecules have entered the clinical trial, and none progressed to phase III. This paper reviews the challenges and opportunities in the development of miRNA-therapeutics.

  1. Line 103. Explain “gain-of-function”.
  2. Lines 150 and 151. Add “and” between “ miR-145-5p” and “CFTR gene”.
  3. Lines 388-390. “The siRNAs exhibit 100% complementarity 388 with the target gene sequence, ……….. hundreds of genes.” This is not clear.
  4. Several miRNA modulations and delivery approaches were discussed. Among these approaches, which ways are more useful than the others?
  5. It is still not very clear the challenges in the development of miRNA-therapeutics from this paper. Compared with siRNAs, miRNA-therapeutics is less developed than siRNA-therapeutics. Add more discussion of the comparison with siRNA-therapeutics.

Author Response

Comment: microRNAs (miRNAs) play an important role in regulating gene expression and disease pathogenesis. miRNA-therapeutics have the potential to be developed. However, less than 20 miRNAs targeting molecules have entered the clinical trial, and none progressed to phase III. This paper reviews the challenges and opportunities in the development of miRNA-therapeutics.

Response: We appreciate the encouraging words and constructive suggestions.

Comment: Line 103. Explain "gain-of-function."

Response: In this context, we meant miR overexpression studies by the "gain-of-function." We recognize that the use of "gain-of-function" creates confusion and thus has been removed from the revised manuscript.

Comment: Lines 150 and 151. Add "and" between "miR-145-5p" and "CFTR gene".

Response: Corrected. Please see line 169 of the revised manuscript.

Comment: Lines 388-390. "The siRNAs exhibit 100% complementarity 388 with the target gene sequence, ……….. hundreds of genes." This is not clear.

Response: By "100% complementarity," we meant Watson-crick sequence complementarity between siRNA or miR with that of the target gene.  Please see lines 374-376 of the revised manuscript.

Comment: Several miRNA modulations and delivery approaches were discussed. Among these approaches, which ways are more useful than the others?

Response: Thank you for bringing this critical point. We have discussed the advantage and disadvantages of different approaches in the revised manuscript.

  1. miR modulation

Deletion of processing enzymes: As 1000s of miRs are expressed, and each has a complex role in gene regulation, the deletion of the miR processing enzyme is unlikely to be utilized for the miR-therapeutics. Please see lines 109-111 of the revised manuscript.

miR-mimics and miR precursors: The miR-mimics increase the levels of miR and can be developed as miR-therapeutics (e.g., MRG-201; designed to mimic the activity of miR-29, MRX34; designed to mimic the activity of miR-34a). Similarly, the miR precursors also increase the levels of mature miR but were traditionally considered transitory intermediates for mature miR. However, recent studies show the regulatory role of precursor miRs in target recognition. Jafari et al. demonstrated a difference in the anticancer effects of miR-34a and pre-mir-34a and attributed these effects to the stable expression of pre-mir-34a and their regulatory roles. Please see lines 117-124 of the revised manuscript.

miR-inhibitors and miR sponge: In contrast to inhibitors, the sponge transgene is often delivered through a virus, and the transcribed mRNA has multiple binding sites for the target miR. This method of miR inhibition offers some advantages over the miR knockout or the use of miR inhibitor for the loss-of-function studies. For example, the miR knockout approach becomes challenging when closely related miRs are encoded from the different loci, or miR precursors are transcribed in clusters. The proximity of the miRs within a cluster makes it challenging to delete one miR without affecting others. As the sponges interact with the mature miR, their effectiveness is unaffected by the multiple miRs with closely related seed sequences or the clustering of miR precursors. Please see lines 151-160 of the revised manuscript.

  1. miR delivery

The selection of the miR delivery system depends on the nature of the miR-therapeutics (enzyme susceptibility), the expression pattern of the target miR (for miR inhibitors) or gene, intended site of delivery, and adverse effect tolerability. The delivery to the central nervous system possesses the additional challenge of crossing the blood-brain barrier. For example: despite the high delivery efficiency of viral vectors, the activation of host immune response is a concern; the lipid-based particles deliver primarily to the liver and the reticuloendothelial system; the large size of lipid-based particles can be advantageous as they escape renal filtration, allowing higher payload. Please see lines 175-184 of the revised manuscript.

Comment: It is still not very clear the challenges in the development of miRNA-therapeutics from this paper. Compared with siRNAs, miRNA-therapeutics is less developed than siRNA-therapeutics. Add more discussion of the comparison with siRNA-therapeutics.

Response: We have added additional detail on the comparison of miR and siRNA-based therapeutics. Please see section 4, lines 365-389 of the revised manuscript.

Reviewer 3 Report

The review by Yahya Momin and colleagues tries to summarize current research in the field of therapeutic ON delivery to cells to mimic or repress miRNA activity. On the one hand, the authors clearly explain the diverse oligonucleotides (ON) and ON modifications available commenting on the potential delivery mechanisms and providing useful information. Nevertheless, the miRNA biology is instead described in a rather superficial manner, with minimal details on the mechanisms of action of miRNAs. Moreover, there are several conceptual errors as well as pieces of information which have been mistakenly reported in an inaccurate and in some cases totally wrong fashion.

Although the authors cite a huge number of papers documenting a variety of possible techniques, the review does not provide any clear take home message, nor it sheds light on the reasons why miR-therapeutics did not succeed (or can be expected to prove successful in the near future). I therefore honestly doubt that in its present form the review can be useful to other researchers, as no innovative point of view, technical solution or successful example of miR-therapeutice is presented.
Overall, data presented by the authors seem to justify skepticism towards this approach, despite the fact that abstract and Discussion argue in favor of the feasibility of miR-therapeutics.

Major points

1) most of the miRNA molecular biology and most notably, details about the mechanism through which miRNAs bind to their targets have been omitted. This is a key point, as the intrinsically poor specificity of the binding of miRNAs to their target (mediated by as little as 7 nt match) underlies the issue of so called "off targets". The paragraph starting at line 393 does not provide any hint, and based on the content of this paragraph, miR therapeutics is totally pointless. This is at the odds with the premises the authors make in the abstract, as well as at the odds with the conclusions of the manuscript. Yet, the authors do not provide any hint as of how off-targeting can be prevented and/or reduced. Either off-targeting is widespread and unavoidable (and hence miRNA therapeutics is pointless) or off-targeting can be managed, but in this latter case the authors should explain how and to what extent, providing examples.

2) line 60: "Under  specific  conditions (e.g., serum starvation and in quiescence) the miRs also initiate the gene translation". This piece of information is critical, as in the healthy body most of the cells will be quiescent. The authors should comment on this as most of the data in the literature on miRNAs were obtained in in vitro models, mainly immortalized proliferating cell lines, a condition in which miRNA are expected to (and actually do) repress protein expression of their target.

3)Apart from a single line in the introduction, the author never mention the catalytic activity of AGO2. This feature of AGO2 is critical in the design of small RNA therapeutics including "sponges".

4) the authors do not mention that miRNAs bind equally well AGO2, AGO1, AGO3 and AGO4, and that all four proteins contribute to miRNA activity, not just AGO2.

5) In the first row of the table on page 6 the authors mention miR-34a delivery by AV vectors, suggesting it can activate immune response. Ref 135, however clearly states that miR-34a is not related to immune stimulation, but rather delivered to HCC cells to exert a pro-apoptotic and anti-proliferative activity. The Interleukin encoded in the same vector is responsible for immune cell activation instead, an effect that in this case is not related to miR-therapeutics at all.

6) In many instances [ref 9, ref 5] the authors refer to reviews, while scholarly, when a single experimental finding is reported, citing the original research paper would be more appropriate.

Minor points:

the human member of the AGO family are  AGO1 AGO2 AGO3 and AGO4. There is no Ago-2 human protein.

The authors make a long dissertation on the use of immune cells to deliver miR-therapeutics. However there is no solid in vivo data to support the feasibilty of such approach. Hence, the entire section is largely speculative, I suggest shortening it and reducing emphasis on this approach.

The Table on page 6 needs a title and a caption

Author Response

Comment: The review by Yahya Momin and colleagues tries to summarize current research in the field of therapeutic ON delivery to cells to mimic or repress miRNA activity. On the one hand, the authors clearly explain the diverse oligonucleotides (ON) and ON modifications available, commenting on the potential delivery mechanisms and providing useful information. Nevertheless, miRNA biology is instead described in a rather superficial manner, with minimal details on the mechanisms of action of miRNAs. Moreover, there are several conceptual errors as well as pieces of information that have been mistakenly reported in an inaccurate and in some cases totally wrong fashion.

Response: Thank you for the constructive suggestions.

Comment: Although the authors cite a huge number of papers documenting a variety of possible techniques, the review does not provide any clear take-home message, nor it shed light on the reasons why miR-therapeutics did not succeed (or can be expected to prove successful in the near future). I therefore honestly doubt that in its present form the review can be useful to other researchers, as no innovative point of view, technical solution or successful example of miR-therapeutics is presented.

Response: We have now described the miR biology in more detail. Please see section 2, "The miR biogenesis and mechanism of action," of the revised manuscript.

Comment: Overall, data presented by the authors seem to justify skepticism towards this approach, despite the fact that abstract and discussion argue in favor of the feasibility of miR-therapeutics.

Response: We regret that the take-home message was not clear in our writing. Shaped by the suggestions of the reviewer, the take-home message is that "Even though the current failures with the miR-therapeutics is discouraging, the miR-therapeutics has potential in managing complex disease conditions; which involve deregulation of multiple signaling pathways." The specific approaches that can be utilized to fuel the development of miR therapeutics include,

  1. a) nuclease-resistant gamma-modified PNA-based anti-miRs;
  2. b) plant-derived miRs;
  3. c) targeted delivery of miR modulators.

Please see section 6, "conclusions" of the revised manuscript.

Comment: Most of the miRNA molecular biology and most notably, details about the mechanism through which miRNAs bind to their targets have been omitted. This is a key point, as the intrinsically poor specificity of the binding of miRNAs to their target (mediated by as little as 7 nt match) underlies the issue of so called "off targets". The paragraph starting at line 393 does not provide any hint, and based on the content of this paragraph, miR therapeutics is totally pointless. This is at odds with the premises the authors make in the abstract and at odds with the conclusions of the manuscript. Yet, the authors do not provide any hint about how off-targeting can be prevented and/or reduced. Either off-targeting is widespread and unavoidable (and hence miRNA therapeutics is pointless) or off-targeting can be managed, but in this latter case, the authors should explain how and to what extent, providing examples.

Response: Thank you for these insightful suggestions. Yes, the miRs have 100 to 1000  targets, and off-targeting is widespread but can be managed. The adverse effects of miR inhibitors are not only because of the off-target effect. The non-specific interaction of negatively charged DNA-based miR inhibitors with proteins leads to non-specific accumulation in tissues and adverse outcomes. Peptide-nucleic acid-based miR-inhibitors can address this concern. Please see section 5.1, " gamma-modified PNA-based miR inhibitors," and section 3.1.3, " miR-inhibitors and miR sponge," lines 138-143, of the revised manuscript.

Comment: line 60: "Under specific conditions (e.g., serum starvation and in quiescence) the miRs also initiate the gene translation". This information is critical, as in a healthy body, most of the cells will be quiescent. The authors should comment on this as most of the data in the literature on miRNAs were obtained in in vitro models, mainly immortalized proliferating cell lines, a condition in which miRNA are expected to (and actually do) repress protein expression of their target.

Response: Yes, in a healthy body, many cells maintain quiescence. We updated our review to include the literature for the role of miRs during quiescence. Please see lines 56-71 of the revised manuscript.

Comment: Apart from a single line in the introduction, the author never mentions the catalytic activity of AGO2. This feature of AGO2 is critical in the design of small RNA therapeutics, including "sponges." The authors do not mention that miRNAs bind AGO2, AGO1, AGO3 and AGO4 equally well and that all four proteins contribute to miRNA activity, not just AGO2.

Response: In mammals, AGO proteins include AGO1-4, and all bind to and contribute to the miR activity. However, only AGO2 has the mRNA cleaving catalytic activity and is therefore known as a slicer. The description is updated. Please see lines 48-56 of the revised manuscript.

Comment: In the first row of the Table on page 6, the authors mention miR-34a delivery by AV vectors, suggesting it can activate immune response. Ref 135, however clearly states that miR-34a is not related to immune stimulation, but rather delivered to HCC cells to exert a pro-apoptotic and anti-proliferative activity. The Interleukin encoded in the same vector is responsible for immune cell activation instead, an effect that is not related to miR-therapeutics.

Response: We regret the error in interpreting the cited paper incorrectly. The Table is not part of the revised manuscript. We have shortened the miR-delivery section and expanded on the mechanism, possible reasons for the high failure of miR medicine, and opportunities. 

Comment: In many instances [ref 9, ref 5] the authors refer to reviews, while scholarly, when a single experimental finding is reported, citing the original research paper would be more appropriate.

Response: The citation of review articles has been kept to a minimal.

Comment: The human member of the AGO family are AGO1, AGO2, AGO3, and AGO4. There is no Ago-2 human protein.

Response: Corrected.

Comment: The authors make a long dissertation on the use of immune cells to deliver miR-therapeutics. However, there is no solid in vivo data to support the feasibility of such an approach. Hence, the entire section is mainly speculative, and I suggest shortening it and reducing the emphasis on this approach.

Response: As suggested, this section is shortened.

Comment: The Table on page 6 needs a title and a caption.

Response: The Table is deleted.

Round 2

Reviewer 1 Report

The review by Yahya Momin and colleagues tries to summarize current research in the field of therapeutic delivery to cells to mimic or repress miRNA activity. The authors clearly explain the diverse oligonucleotides  and  modifications available, commenting on the mechanisms and providing useful information. Nevertheless, miRNA biology is instead described in a rather superficial manner, with minimal details on the mechanisms of action of miRNAs. I am accordingly with the reviewers that this paper shoul be completely revised .

Author Response

Comment: The review by Yahya Momin and colleagues tries to summarize current research in the field of therapeutic delivery to cells to mimic or repress miRNA activity. The authors clearly explain the diverse oligonucleotides and modifications available, commenting on the mechanisms and providing useful information. Nevertheless, miRNA biology is instead described in a rather superficial manner, with minimal details on the mechanisms of action of miRNAs. I am accordingly with the reviewers that this paper should be completely revised.

Response: We are happy to learn that the reviewer likes the explanation we provided on the diverse oligonucleotides and modifications available and our comments on the mechanisms. Considering the focus of this review (challenges and opportunities in the development of miR-therapeutics), we limited our discussion on the "biology of miRNA and mechanisms of action" in the original submission. Nonetheless, as per the reviewer's suggestions, we added additional discussion on the "biology of miRNA and mechanisms of action." Please see lines 45-68, 76-88, and 105-109 of the revised manuscript. We have also extensively revised this manuscript (which includes; refining the miR delivery section, adding more details in the miR biology and mechanism of action section, and adding details in the opportunities section). Besides, we corrected the English of the entire manuscript to improve readability.

Reviewer 3 Report

I have carefully read replies by the authors to my comments as well as the updated text of the review. My major concern however is still unaddressed: the review reports on failure of existing miRNA therapeutics trials, yet conclusions argue in favour of this technique. However, based on my understanding of the text, the authors do not provide sufficient evidences that alternative approaches may actually be succesfully translated to the clinic, as for all the potential approaches they mention, research is still at very basic level, with data stemming from computational analyses or in vitro study, and no pre-clinical data are available.

Although the authors integrated some of the missing information, I still believe that the opinion by the authors of future possibilities in the field of miRNA therapeutics is not supported by data and facts, but relies instead on a small set of preliminary observations and is largely speculative.

Gamma-modified PNA-based miR inhibitors: the authors suggest that this new class of molecules might potentially overcome the limits of other classes of inihibitors. While the features of Gamma PNA may be beneficial, there are also concerns associated with their in vivo use, mainly due to their ability to interact with DNA and promote recombination. The study (ref 139) by the authors is certainly relevant. However to the best of my knowledge, that study is focused on the efficacy of the gamma PNA in the repression of miR-155 (as compared to other PNA ON). In ref 139 the safety, pharmacokinetics and other relevant issues which are mandatory before proceeding to clinical trials have not been addressed. Neither those have been addressed in other publications (at least based on references provided by the authors). While the data of ref 139 are certainly sufficient to suggest that Gamma PNA miR inhibitors should be further investigated (i.e. testing inhibition of other miRNAs, testing safety, genotoxicity pharmacokinetics in animal models), this only report  does not substantiate (in my opinion) the claim that "employing γPNA technology represents a novel approach to develop miR-therapeutics"

5.2. Plant-derived miRs: I disagree with the view that plant derived miRNAs are a case of miRNA therapeutics, as in this case we are not speaking of upregulating od inhibiting miRNAs into host cells, but rather discussing the potential of RNA molecules which might happen to regulate host microbiome. Also in this case, conclusions by the authors are drawn mainly by (over)interpretation of a single paper (and a set of previous reports by the same group) supporting the idea that ingested plant miRNA are uptaken by stomach cells and regulate expression of host genes. Although this is certainly an intriguing finding, the authors apparently disregard the huge debate in the scientific community about orally administered plant miRNAs (reviewed in PMID: 31193509). The authors should discuss these issues to provide an unbiased view to the reader.

Author Response

Reviewer 3

Comment: I have carefully read replies by the authors to my comments and the updated text of the review. My major concern however is still unaddressed: the review reports on failure of existing miRNA therapeutics trials, yet conclusions argue in favor of this technique. However, based on my understanding of the text, the authors do not provide sufficient evidence that alternative approaches may actually be successfully translated to the clinic, as for all the potential approaches they mention, research is still at very basic level, with data stemming from computational analyses or in vitro study, and no preclinical data are available. Although the authors integrated some of the missing information, I still believe that the opinion by the authors of future possibilities in the field of miRNA therapeutics is not supported by data and facts but relies instead on a small set of preliminary observations and is largely speculative.

(i) Gamma-modified PNA-based miR inhibitors: the authors suggest that this new class of molecules might potentially overcome the limits of other classes of inhibitors. While the features of Gamma PNA may be beneficial, there are also concerns associated with their in vivo use, mainly due to their ability to interact with DNA and promote recombination. The study (ref 139) by the authors is certainly relevant. However to the best of my knowledge, that study is focused on the efficacy of the gamma PNA in the repression of miR-155 (as compared to other PNA ON). In ref 139 the safety, pharmacokinetics and other relevant issues which are mandatory before proceeding to clinical trials have not been addressed. Neither those have been addressed in other publications (at least based on references provided by the authors). While the data of ref 139 are certainly sufficient to suggest that Gamma PNA miR inhibitors should be further investigated (i.e. testing inhibition of other miRNAs, testing safety, genotoxicity pharmacokinetics in animal models), this only report does not substantiate (in my opinion) the claim that "employing γPNA technology represents a novel approach to develop miR-therapeutics."

(ii) Plant-derived miRs: I disagree with the view that plant-derived miRNAs are a case of miRNA therapeutics; as in this case, we are not speaking of upregulating or inhibiting miRNAs into host cells, but rather discussing the potential of RNA molecules which might happen to regulate host microbiome. Also, in this case, conclusions by the authors are drawn mainly by (over)interpretation of a single paper (and a set of previous reports by the same group) supporting the idea that ingested plant miRNA are uptaken by stomach cells and regulate expression of host genes. Although this is certainly an intriguing finding, the authors apparently disregard the huge debate in the scientific community about orally administered plant miRNAs (reviewed in PMID: 31193509). The authors should discuss these issues to provide an unbiased view to the reader.

Response: Thank you for the critical suggestions. We understand the reviewer's primary concern "review reports on the failure of existing miRNA therapeutics trials, yet conclusions argue in favor of this technique." We report the failure of existing miR therapeutics to highlight the current developments. In conclusion, we argued in favor of miR-therapeutics based on the novel opportunities, in which we discussed gamma PNA technology and plant-derived miRs.

γPNAs: The reviewer pointed out that only one study (ref# 139 in the previous version, ref # 154 in the revised manuscript) showed efficacy in the animal-based study. Safety, pharmacokinetics, and other relevant mandatory issues before proceeding to clinical trials have not been addressed.

In the revised manuscript, we provide: (a) 6 reports that show the efficacy of gamma PNA technology in preclinical studies (Ref # 28, 150-154), (b) 4 reports that show the biocompatibility of γPNA technology (ref#  28, 150, 153 and 155), and (c) 2 reports that show the pharmacokinetics of γPNAs (ref # 127 and 163). We agree that γPNA technology is in the early stages of development, and data is not exhaustive. However, based on existing evidence, we believe that this technology can generate next-generation miR inhibitors that are more efficient and safer. Please see lines 527-553 of the revised manuscript.

Plant-derived miRs: Thank you for directing our attention to the scientific debate on plant-derived miRs. Plant-derived miRs regulate (i) host intestinal microbiota (ref # 164), (ii) gene expression in host cells (miR168a: SIRT1, ref # 168, miR159: TCF7, ref # 165), and (iii) disease progression in the host (ref # 165 and 170). Besides, the plasma levels of miRs inversely correlate with cancer's incidence and progression (ref # 165). Therefore, we included them as potential miR therapeutics.

As suggested, we have included more details on the plant-derived miRs to support their role as potential therapeutics and discussed the possible loopholes of this approach to provide an unbiased view to the reader. Please see lines 555-613 of the revised manuscript.